# On the Crashworthiness Behaviour of Innovative Sandwich Shock Absorbers

**DOI:** 10.3390/polym14194163

**Published:** 2022-10-04

**Authors:** Valerio Acanfora, Ferdinando Baldieri, Antonio Garofano, Francesco Fittipaldi, Aniello Riccio

**Affiliations:** Department of Engineering, University of Campania “L. Vanvitelli”, via Roma, 29, 81031 Aversa, Italy

**Keywords:** composite materials, crashworthiness, additive manufacturing, hybrid shock absorber, finite elements

## Abstract

Increasing the impact resistance properties of any transport vehicle is a real engineering challenge. This challenge is addressed in this paper by proposing a high-performing structural solution. Hence, the performance, in terms of improvement of the energy absorbing characteristics and the reduction of the peak accelerations, of highly efficient shock absorbers integrated in key locations of a minibus chassis have been assessed by means of numerical crash simulations. The high efficiency of the proposed damping system has been achieved by improving the current design and manufacturing process of the state-of-the-art shock absorbers. Indeed, the proposed passive safety system is composed of additive manufactured, hybrid polymer/composite (Polypropylene/Composite Fibres Reinforced Polymers—PP/CFRP) shock absorbers. The resulting hybrid component combines the high stiffness-to-mass and strength-to-mass ratios characteristic of the composites with the capability of the PP to dissipate energy by plastic deformation. Moreover, thanks to the Additive Manufacturing (AM) technique, low-mass and low-volume highly-efficient shock-absorbing sandwich structures can be designed and manufactured. The use of high-efficiency additively manufactured sandwich shock absorbers has been demonstrated as an effective way to improve the passive safety of passengers, achieving a reduction in the peak of the reaction force and energy absorbed in the safety cage of the chassis’ structure, respectively, up to up to 30 kN and 25%.

## 1. Introduction

Nowadays, crashworthiness properties are a widely explored topic in the design of safe and effective transportation systems [1,2,3,4]. The development of concepts aimed to improve the passive safety of transport vehicles is one of the most popular challenges in contemporary engineering. Among the proposed solutions, a significant contribution to this issue was reached by introducing shock absorbers in strategic structure locations [5,6,7,8,9]. In particular, shock absorbers have been found to be a good alternative to the high-pressure chamber landing system usually installed on helicopters, showing appreciable energy absorbing capabilities during crash events [5]. In [6], the effectiveness of hexagonal metal honeycomb structures has been investigated as crashworthy structures in a lunar lander system. In [7], crushable hybrid energy absorbers have been incorporated as vertical struts in a fuselage structure to improve the crashworthiness of aircrafts. In [8], the employment of energy absorbing devices incorporated in the seats and subfloor section of a helicopter has been investigated as an effective way to reduce the impact loads transferred to passengers. Furthermore, the increase in the shock absorber’s energy damping capability, resulting from the structural combination of ductile and composite materials, is demonstrated in [9], where a hybrid metal/composite shock absorber system was placed in the bottom of an ejection seat, absorbing more than 50% of the resulting energy. Sandwich structures represent an optimum solution among the different available shock absorber structures thanks to their remarkable energy absorbing capability.

Sandwich shock absorbers are usually composed of ductile materials, which allow the dissipation of high energy rates through plasticisation [10,11,12]. Nevertheless, such solutions are characterised by a significant footprint in terms of mass and volume [13]. Nowadays, innovative production processes based on additive manufacturing help to overcome this limitation [9,11,14], allowing structural lightening by adopting complex shapes (e.g., latex domains), while preserving energy absorption capabilities [15].

An applicative test case in the automotive field of this new structural shock absorbers’ concept is proposed in this work. In particular, this work aims to quantify the benefit of introducing several shock absorber systems, designed for Additive Manufacturing (DfAM) in key locations of a minibus chassis.

The dynamic response of the introduced absorber configurations has already been assessed both numerically and experimentally in [15,16]. Indeed, the absorber geometry is the same reported in [15]. In particular, it consists of a sandwich configuration with Polypropylene (PP) core and internal skin and carbon fibre composite (CFRP) external coating. This allows to combine the flexibility of the polymeric component with the lightness and strength of the external composite skins. Moreover, the selection of PP was also made to fulfil the European Commission’s directives in terms of recyclability. Indeed, as polypropylene is a thermoplastic polymer, it is totally recyclable.

All components in the shock absorber device fulfil a specific role during the impact phenomena: the polypropylene core can easily dissipate the impact energy through plasticisation mechanisms, and the composite skin reinforces the sandwich structure and contributes to the energy dissipation through onset and propagation of intralaminar damages, while the PP skin allows a better adhesion between the PP core and the composite skin, avoiding delamination. The employment of a thinner and a thicker core allows the investigation of the influence of the PP core thickness on the energy absorption capabilities of the device. The mathematical chassis and shock absorber discretization has been performed in the Abaqus environment adopting a detailed approach that adopts 3D elements formulation (C3D8R and SC8R).

The investigation has been conducted by comparing the results, in terms of stress field, deformations, and energy recorded in the safety cage of the chassis, resulting from numerical crash test simulations performed on the steel frame of the standard configuration and of ones equipped with different hybrid shock absorber systems.

In accordance with the Euro NCAP standard for full frontal crash tests, the impact speed considered in the explicit simulation is 50 km/h. However, this is the maximum achievable impact velocity (due to design constraints) for the minibus.

A description of the mathematical assumption and materials adopted to perform the numerical investigation is introduced in Section 2. The developed FE models are presented in Section 3, and a comparison and discussion of the obtained numerical results is provided in Section 4. Finally, a detailed theoretical background has been presented in Appendix A.

## 2. Theoretical Remarks and Materials

The numerical crash test simulations have been performed by means of dynamic explicit analyses in the Abaqus environment. The failure mechanisms of the considered materials have been taken into account in the numerical analysis as well. In particular, the onset and evolution of the intralaminar damage mechanisms of the shock absorber’s composite skins component has been assessed by using the Hashin’s criteria, based on a Continuous Damage Mechanics (CDM) formulation; this approach allows one to identify the breakage of fibres and matrix and investigate the evolution of the damages under tensile and compressive loading. Moreover, the elasto-plastic behaviour of the polymeric component has been evaluated by means of its σ-ε curve; finally, the ductile damage mechanisms of the A-36 steel component, which constitutes the chassis’ structure, has been considered by introducing the Johnson–Cook criteria.

A detailed theoretical background on the explicit integration scheme for the crash simulations and on the damage mechanisms for steel, composite and polymeric materials employed in this work are briefly reported in Appendix A.

The composite material considered in this investigation is IM7/977-2 carbon fibre-reinforced polymer composite. Its mechanical properties, implemented in the numerical software, are reported in Table 1. The main mechanical properties of the polymeric PP material, involved as an internal component of the shock absorbers, are listed in Table 2, while the mechanical properties of the ASTM A36 steel (both elastic and ductile J-C damage properties) considered as material of the chassis structure are given in Table 3.

## 3. FE Model

In this section, information on the geometry of the structures, the discretisation of the finite element models, and the set-up imposed for the numerical crash test investigation are provided.

In particular, Section 3.1 introduces the geometrical characteristics of the introduced configurations, Section 3.2 presents a sensitivity analysis of the computational grid aimed to select a suitable finite element dimension, and, finally, Section 3.3 describes the BCs used to simulate the numerical crash tests.

### 3.1. Structure Geometry

By adding to the basic chassis four different shock-absorbing systems, five different configurations have been defined. Figure 1 displays the arrangement of the shock absorbers on the front of the chassis.

In detail:

Chassis 1 identifies the all-steel structure, without shock absorbers (Figure 1A);

In chassis 2, a system composed of two shock absorbers with a total sandwich height of 15 mm has been introduced. The mass of this damping system is 0.264 kg (Figure 1B);

In chassis 3, the shock absorption system of chassis 2 has been extended, assuming a total of six devices placed in the corners and in the middle of the structure’s frontal section, each one with a sandwich height of 15 mm. The total shock absorbing system mass is 0.792 kg (Figure 1C);

Chassis 4 includes a damping system with the same mass as the one introduced in Chassis 3, but composed of only four absorbers. In this configuration, the shock absorbers are placed only in the corners of the structure’s frontal section. However, each absorber is 23 mm high. The mass of this damping system is 0.792 kg (Figure 1D).

This type of configuration with four shock absorbers was considered because the composite amount is reduced by approximately 33% compared to the Chassis 3 configuration (as it has eight rather than twelve composite skins). In addition, since it is non-recyclable and the most expensive part of the shock absorber, this makes this solution cheaper and more environmentally friendly than the Chassis 3 one.

Moreover, in order to further reduce the masses (and consequently the costs) of the proposed solutions, the effectiveness of another solution (Chassis 5) with four shock absorbers with the same thin core structure as Chassis 3, and placed at the same critical points as Chassis 4, is explored. The total mass of this shock absorption system is 0.528 kg (Figure 1E).

Figure 2 shows the geometrical dimensions of the chassis, which is the part shared by each configuration, in different views. According to Figure 2, the longitudinal chassis dimension (in the X direction) is 5375 mm. This length classifies this structure as representative of a minibus chassis.

Figure 3 introduces the geometry of the shock absorbers. It should be noted that the height of the core is indicated by a parameter labelled T, which assumes two different values according to the different configuration: 7 mm for the system in Chassis 2, Chassis 3, and Chassis 5, and 15 mm for the system in Chassis 4. The thickness of each PP and CFRP skin is 2 mm.

As expressed in Figure 3, the shock-absorbing device has been obtained by combining two materials: polypropylene (PP Roboze—red region) and 16 CFRP Composite layers (IM7/977-2—ivory region), each with a thickness of 0.125 mm and stacked according the sequence [0; 45; −45; 90]_s2_, have been employed as external skins.

As demonstrated in the literature [15], this approach is particularly advantageous for structures relegated to the dissipation of impact energy as it allows for the combination of the ability to absorb energy in plastic form by the polymeric core with the high stiffness-to-weight and strength-to-weight ratios of composite external skins.

Moreover, the extremely thin wall of the honeycomb should also be noted. Indeed, in the frame of additive manufacturing, it has been set to 1 mm. This confers a high flexibility to the structure, maximising its energy absorption capability.

Finally, in order to display the interaction between the chassis and the damping systems, an enlargement of the section of the chassis where the latter are placed is provided in Figure 4.

### 3.2. Sensitivity Analysis

The mathematical approach employed for the development of FE models is based on a detailed formulation with three-dimensional finite elements. In particular, as indicated in Figure 5, the chassis structure was discretised using SC8R elements (coloured in green), while, for the shock absorbers, C3D8R (coloured in grey) elements were selected for the polymer component and SC8R elements for the external composite skins.

The size of the FE was defined assessing the solution accuracy related to the computational cost by performing linear static simulations, imposing a 5 mm traction displacement to the front wheel axle and locking the rear wheel axle.

Figure 6 displays the applied load conditions, while Figure 7 compares the different configurations resulting from the variation of the FE size, in terms of von Mises Sigma (Figure 7A) and computational time (Figure 7B), normalized with respect to the configuration with maximum computational time.

The mesh sensitivity analysis proves that a discretization of the chassis with elements smaller than 10 mm does not provide an improvement in the solution, since from this element size the stress stabilises at 127 MPa. However, as shown in Figure 7B, there is a significant increase in computational time when the chassis is discretized with elements smaller than 10 mm. Therefore, 10 mm element size dimension has been selected to discretize the chassis. This finite element dimension overcomes the hourglass problems to which these reduced-integration elements are subject.

Following this preliminary numerical investigation, the definitive FE model of the chassis was composed of 4901200 SC8R elements, while 1039040 C3D8R elements and 480000 SC8R elements formed each shock absorber.

### 3.3. Crash Test Set-Up

Numerically modelled road and rigid wall have been added to perform the full-frontal crash test analyses. The road has been modelled as a 7000 × 3000 mm^2^ rectangular planar shell, discretised by using S4R elements. The rigid wall has been modelled as a 3000 × 2500 × 100 mm^3^ solid part, discretised by using C3D8R elements. Both the road and the rigid wall have been considered as rigid bodies by creating two reference points, where the boundary conditions for each region have been assigned. Figure 8 shows the assembly of the crash test set-up.

To perform the crash test analyses, the reference points of both the road and the rigid wall have been clamped, while a 50 km/h initial velocity has been assigned to the chassis in the x-positive direction, according to the full-frontal crash test standard of Euro NCAP and to the maximum speed achievable by the minibus due to design constraints. Moreover, a gravity load is also defined by assigning the gravitational acceleration in the vertical direction to the whole model and considering the density associated with the materials, in order to take into account the gravitational effect as well.

## 4. Numerical Results

The benefit of introducing composite shock-absorption systems at the front rear of the chassis is discussed in this section. In particular, this has been assessed by means of cross-comparison between the data resulting from all the configurations, in terms of von Mises stresses (Figure 9 and Figure 10) and energy recorded in the chassis’ safety cage. Therefore, the comparison in terms of von Mises Sigma highlights that the maximum value of the stress field is reached in the configuration without composites shock absorbers system. Indeed, as detailed in Figure 9, in the Chassis 1 configuration the maximum predicted stress is 794.6 MPa.

A contribution in terms of load dissipation can be found by adding the damping system consisting of just two shock absorbers of the Chassis 2 configuration. Indeed, this configuration is characterized by the maximum stress of 720.0 MPa.

A significant effect was recorded in Chassis 3, Chassis 4, and Chassis 5 configurations. Indeed, by introducing six shock absorbers (Chassis 3), the maximum stress reached is 652.9 MPa, whereas by introducing damping systems with four shock absorbers, the stress field reaches 670.6 MPa in Chassis 5 and the minimum value of 604.1 MPa in Chassis 4.

In all structural responses, the stress field exceeds the allowable stress limit value for this test category, which is equal to 400 MPa. However, as shown in Figure 10, it can be seen that integrating in the chassis’ structure the damping systems, the area characterized by values exceeding the maximum limit of 400 MPa significantly reduces.

It should be noted that in Chassis 3, Chassis 4, and Chassis 5 configurations, these particularly critical areas are limited to a few points where there are multiple frame intersections. Hence, in these cases, this effect could be easily solved by installing joints which, compared to welding, provide a less rigid connection.

An enlargement on the chassis-shock absorbing systems interaction region, after the crash simulations, is reported in Figure 11.

Figure 12 compares the percentage of total energy transferred in the safety cage (which is the most critical area of the frame) between the explored configurations. This graph shows that by integrating shock-absorbing systems at specific points in the chassis, it is possible to reduce the percentage of impact energy transferred to the safety cage up to 25%. Specifically, for the Chassis 1 configuration, the total energy transferred in the safety cage is 90%, for Chassis 2 it is 81%, for Chassis 3 it is 78%, while in Chassis 4 and Chassis 5 configurations it is 75% and 80%, respectively.

It should be noted that the largest energy gap is observed when comparing the responses of the undamped configuration (Chassis 1) with that of Chassis 2, which is the configuration with the minimum number of absorbers. This means that shock absorbers help to dissipate impact energy immediately. This means that the shock absorbers contribute effectively to dissipating the impact energy.

From an energy point of view, the shock-absorbing systems also introduce another important effect, which can be deduced from the force–time charts comparison shown in Figure 13.

Indeed, by comparing the curves on the force–time graph, it can be seen that by introducing the shock-absorbing systems, the peak force decreases. This is most evident when comparing the undamped Chassis 1 configuration with the Chassis 3 and Chassis 4 configurations, where the peak force is reduced by approximately 30 kN. Moreover, it should be also noted that in the first interval time, the blue curve of the Chassis 1 configuration exhibits a sudden collapse. This is because there is no damping and the phenomenon immediately triggers the catastrophic failure of the structure. This is prevented equipping the damping systems. Indeed, all shock absorber reinforced arrangements assume a horizontal trend, since the damping systems confers to the assembly the capability to dissipate impact energy by means of plastic deformation of the shock absorbers’ core and intralaminar cracking of the composite skins. These effects (maximum peak force reduction and peaks damping in horizontal trend) significantly improve the crashworthiness of the structure, resulting, in practical terms, in a reduced perception of impact acceleration by the passengers and therefore an increase in their safety and comfort.

The energy absorbing contributions of shock absorbers through core plasticisation and composite intralaminar damage assessed according to Hashin’s failure indices are depicted in Figure 14 and Figure 15, respectively.

By comparing the data shown in Figure 14, it is evident that the crash test has a greater effect on plastic deformations in the Chassis 4 and Chassis 5 configurations. The lowest values are recorded in the Chassis 3 configuration. This could induce one to think erroneously that the Chassis 2 shock absorber system is more efficient in terms of absorption and dissipation of impact energy than the Chassis 3 one. However, this is not true since the plastic contribution provided by the shock absorber system is also a function of the number of absorbers of which it is composed. Indeed, the Chassis 2 shock absorber system dissipates more energy through plastic deformation than that of Chassis 3 only from a local point of view.

Among all configurations, the one that dissipates the maximum amount of energy in plastic form in the single absorber is Chassis 4. This is due to the fact that in this configuration, each absorber has a higher proportion of thermoplastic polymer in its core than in the one of the others arrangements.

Finally, as deducible from Figure 12 and Figure 13, due to the greater number of shock absorbers, the total amount of energy dissipated by the Chassis 3 system is higher respect to the Chassis 2 system.

In Figure 15, a comparison of the intralaminar damages in the upper skin of a representative absorber of each shock-absorbing system is presented.

The adopted failure criteria allowed for the investigation of the specific intralaminar failure modes of the composite components.

The greatest amount of intralaminar damages occurs in Chassis 2 and Chassis 5 configurations. Indeed, by comparing the shock-absorbing systems of these two configurations with the one present in Chassis 3, it is evident that in these two cases the impact loads are distributed over a smaller surface area before being propagated over the chassis. This is because Chassis 3’s shock-absorbing system is composed of the maximum number of six shock absorbers, while in these two cases the absorbers are two and four, respectively. On the other hand, conducting the same comparison with Chassis 4, it appears that the intralaminar damages of Chassis 2 and Chassis 5 (but also Chassis 3) are greater. This can be explained because in Chassis 4’s set-up, there is a greater thickness of the plastic core that induces the dissipation of the impact energy mainly through plastic deformations.

In Figure 16, the behaviour of a representative absorber of each shock-absorbing system is compared in terms of deformed shape. The comparison allowed the investigation of the deformation values in the X direction, which is the direction of the crash test, in order to better understand the influence and the contribution of the plastic core in the energy absorption capability of the different shock absorber configurations.

It should be noted that in the configuration with the largest amount of plastic material, i.e., Chassis 4, the structure assumes the maximum deformation values in the X direction (the direction of the crash test), which is equal to 9.60 mm. Meanwhile, in Chassis 2, Chassis 3, and Chassis 5, the maximum deformation is 4.65 mm, 3.89 mm, and 6.02 mm, respectively. However, in terms of intralaminar damage mechanisms, the Chassis 4 configuration shows smaller extensions in terms of both fibre and matrix breakages. This suggests that this type of damping configuration has a lower stiffness than the other ones, but, at the same time, it is the most efficient in terms of energy absorption. On the other hand, if we measure the ratio between the total absorbed energy of the shock-absorbing system and its mass (SEA index, Specific Energy Absorption [15]), it can be seen that the most efficient setups are the ones of Chassis 2 and Chassis 5. Therefore, it can be affirmed that it is not easy to identify a solution that is the most efficient one, because each of them exhibits specific advantages, and the choice of the configuration should be based on the specific design requirements. Finally, a summary table (Table 4) of the characteristics that emerged from each configuration is provided in order to allow an easy trade-off between the results of this analysis.

## 5. Conclusions

This paper assesses the effectiveness, in terms of energy dissipation, of PP/CFRP composite shock absorbers designed for Additive Manufacturing, under realistic operating conditions.

In particular, the effects on the maximum von Mises stress reached on the chassis of minibuses subjected to a 50 km/h crash test, with and without the integration of different damping systems, are compared. The work shows that the introduction of dampers can reduce the maximum stress on the chassis by up to 24%. Despite in all damping system configurations the stress fields exceed the allowable stress limit value, the employment of the shock absorber devices is able to significantly reduce the area characterized by values exceeding the maximum limit. Furthermore, the effectiveness of the energy dissipation systems has been assessed by comparing the energy recorded in the safety cage, the force–time graphs, and the damage mechanisms of the composite shock absorbers.

The comparison of the energy recorded in the safety cage demonstrated that the introduction of the shock absorber devices can reduce the energy transferred in the safety cage up to 25%. The performed analyses showed that the 90% of the total energy was transferred to the safety cage in the case without shock absorber devices, while the employment of only two shock absorbers in Chassis 2 configuration demonstrated the ability to reduce the total transferred energy to 81%. A maximum reduction in the total energy transferred to the safety cage to 75% has been achieved in the Chassis 4 configuration by means of four thicker devices.

Moreover, the comparison of the curves on the force–time graph showed that the introduction of the shock absorber devices allowed a reduction in the maximum peak of the reaction force up to 30 kN. The ability of the damping systems to dissipate impact energy by means of plastic deformation of the shock absorbers’ core and intralaminar cracking of the composite skins significantly improves the crashworthiness of the structure and passive safety of passengers, resulting in a reduced perception of impact acceleration by the occupants. The comparisons of the equivalent plastic strains for the shock absorbers’ cores and of the intralaminar damages for the composite skins demonstrated that both components effectively contribute to the dissipation of the impact energy.

## Figures and Tables

**Figure 1 polymers-14-04163-f001:**
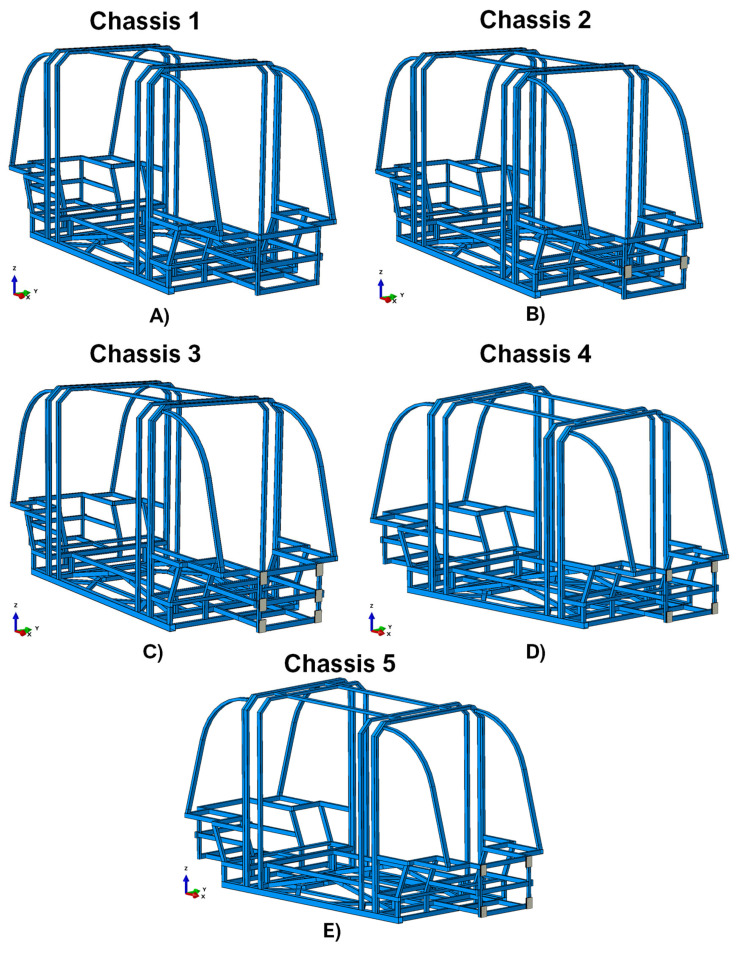
Explored chassis configurations. (**A**) Configuration not equipped with damping systems (Chassis 1); (**B**) Configuration with a damping system composed by two shock absorbers (Chassis 2); (**C**) Configuration with a damping system composed by six shock absorbers (Chassis 3); (**D**) Configuration with a damping system consisting of four shock absorbers with a higher polymer core than the other ones (Chassis 4); (**E**) Configuration with a damping system consisting of four shock absorbers with a thin polymer core (as the one in Chassis 3).

**Figure 2 polymers-14-04163-f002:**
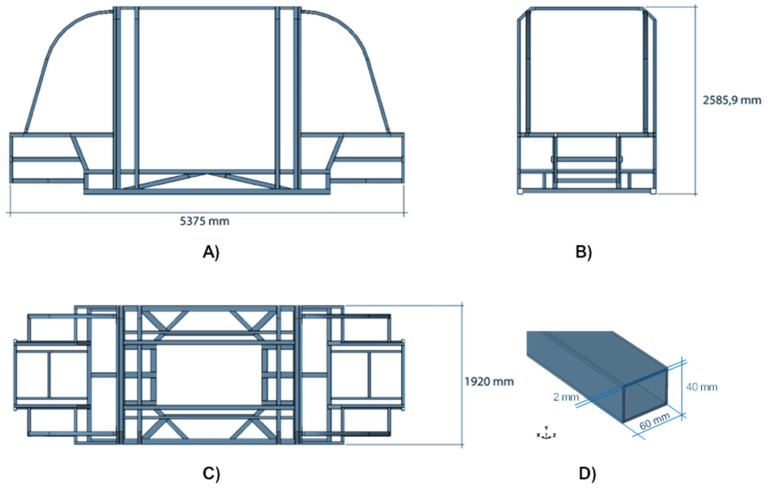
Geometrical dimensions of the chassis. (**A**) lateral view, (**B**) frontal view, (**C**) top view, and (**D**) detail of the frame component.

**Figure 3 polymers-14-04163-f003:**
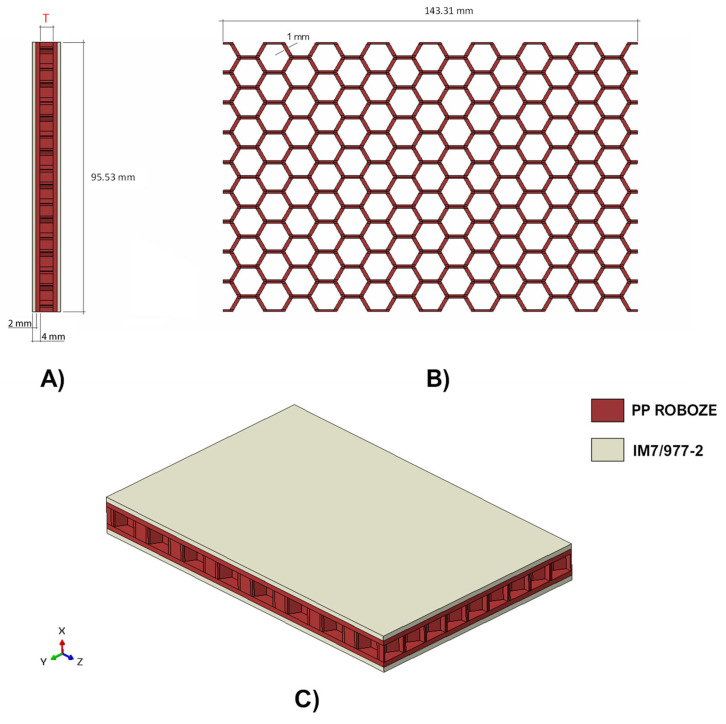
Geometrical details of the shock absorber. (**A**) lateral view, (**B**) top view, and (**C**) isometric view.

**Figure 4 polymers-14-04163-f004:**
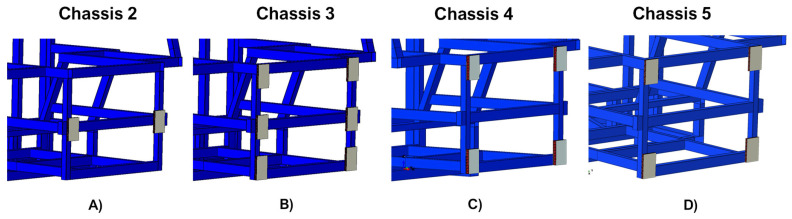
Chassis-damping systems interaction zone. (**A**) Chassis 2, (**B**) Chassis 3, (**C**) Chassis 4, and (**D**) Chassis 5.

**Figure 5 polymers-14-04163-f005:**
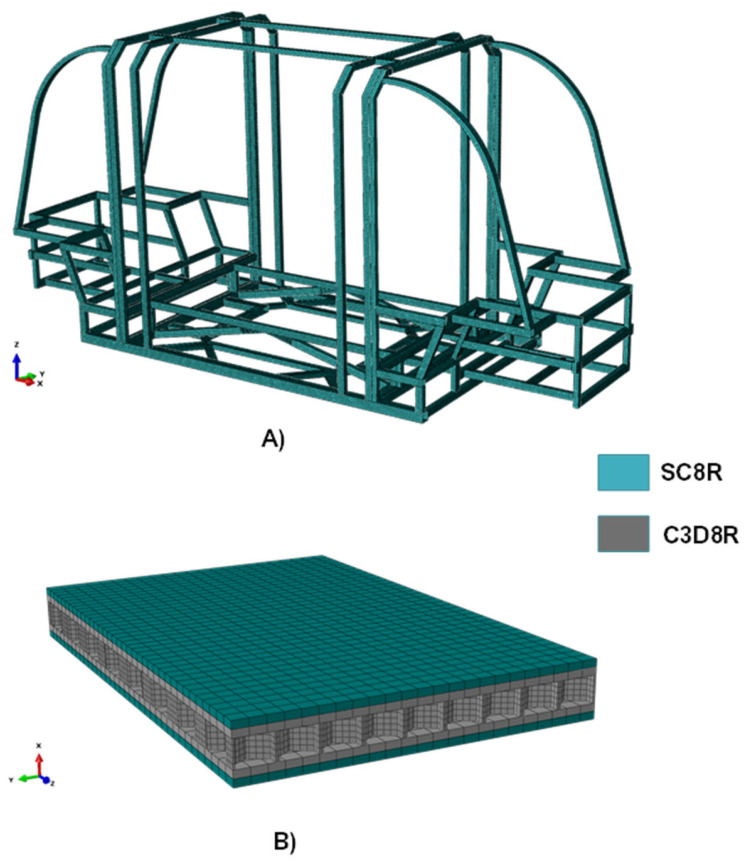
Finite element discretization. (**A**) Computational grid of the chassis with SC8R elements, (**B**) Computational grid of the shock absorber with SC8R elements (in green) and C3D8R elements (in grey).

**Figure 6 polymers-14-04163-f006:**
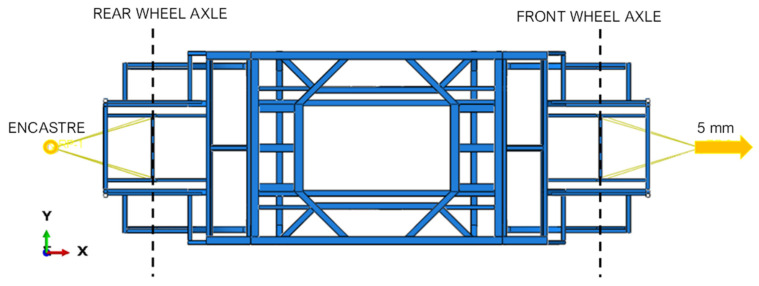
Finite Element discretization.

**Figure 7 polymers-14-04163-f007:**
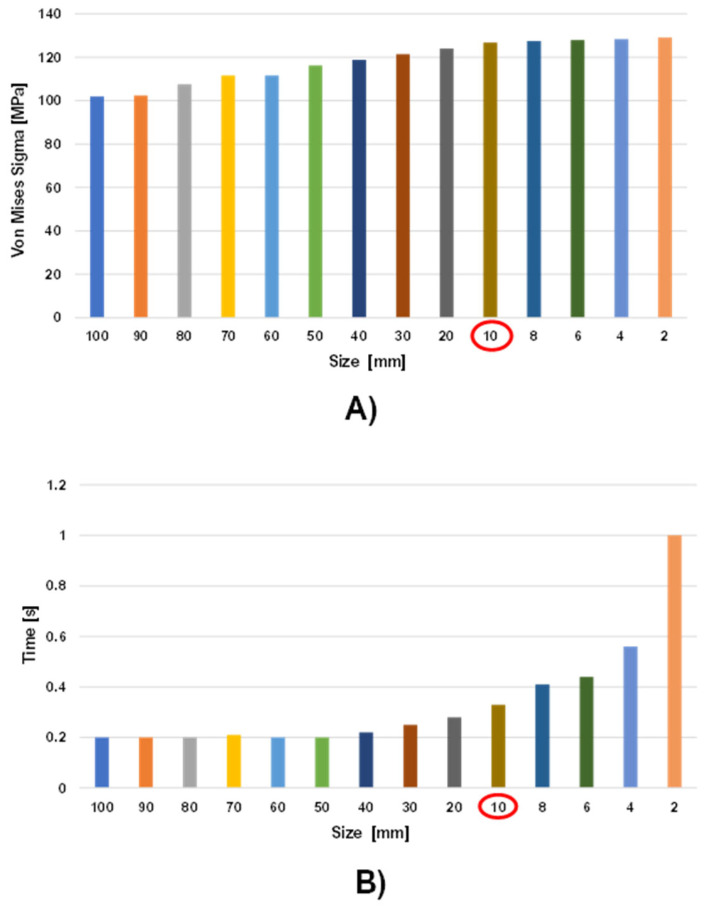
Finite element discretization. (**A**) Histogram graph of stresses as a function of element size; (**B**) Histogram graph of calculation time as a function of element size.

**Figure 8 polymers-14-04163-f008:**
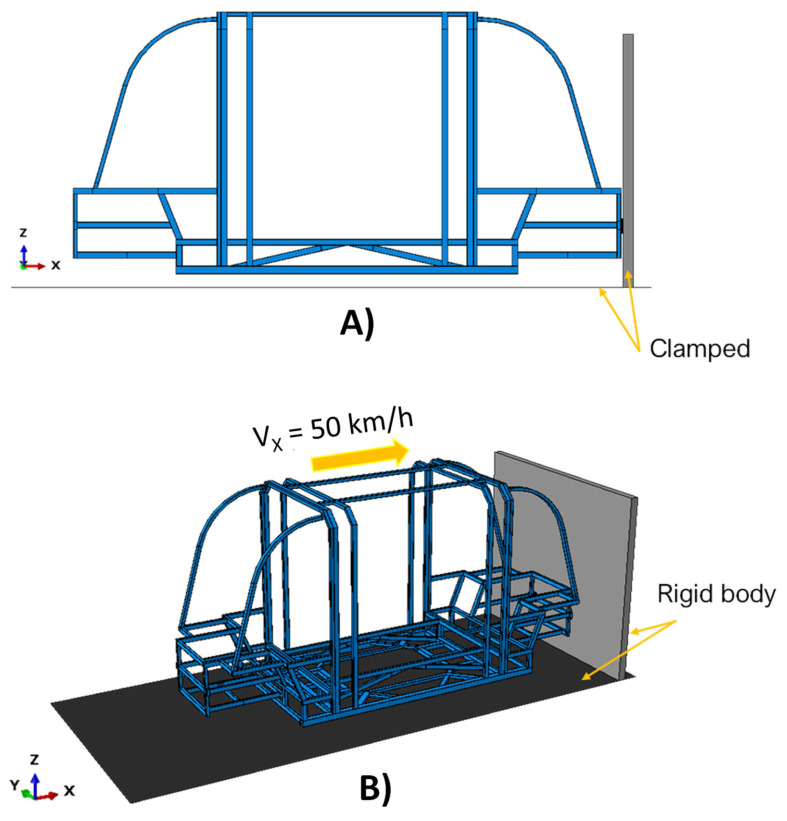
Crash test set-up. (**A**) lateral view, (**B**) isometric view.

**Figure 9 polymers-14-04163-f009:**
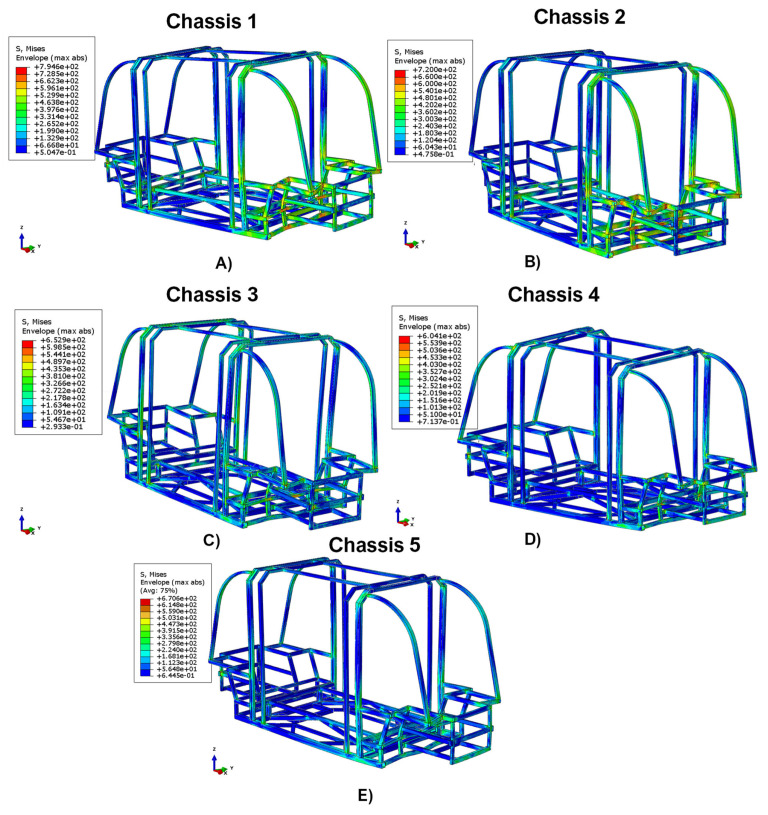
Comparison in terms of Von Mises’s stress. (**A**) Chassis 1, (**B**) Chassis 2, (**C**) Chassis 3, (**D**) Chassis 4, and (**E**) Chassis 5.

**Figure 10 polymers-14-04163-f010:**
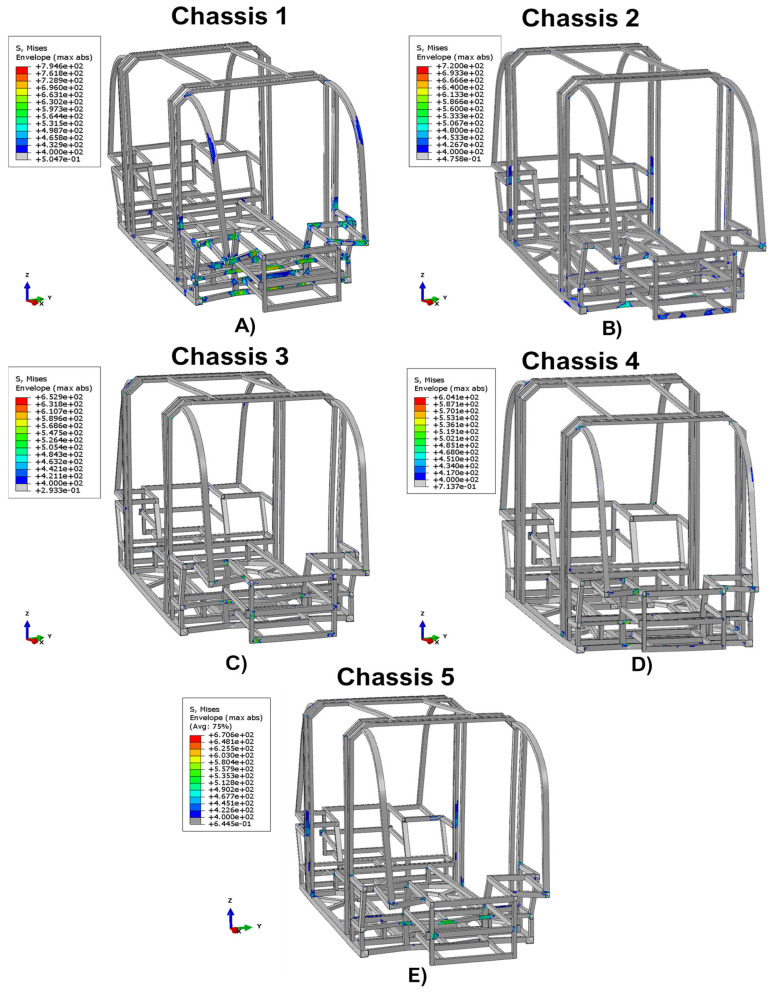
Comparison with stress exceeding 400 MPa. (**A**) Chassis 1, (**B**) Chassis 2, (**C**) Chassis 3, (**D**) Chassis 4, and (**E**) Chassis 5.

**Figure 11 polymers-14-04163-f011:**
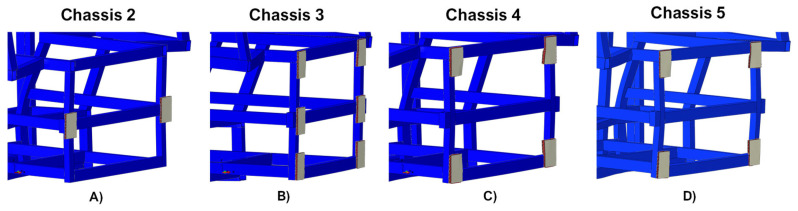
Chassis-damping systems interaction zone after crash tests. (**A**) Chassis 2; (**B**) Chassis 3; (**C**) Chassis 4; (**D**) Chassis 5.

**Figure 12 polymers-14-04163-f012:**
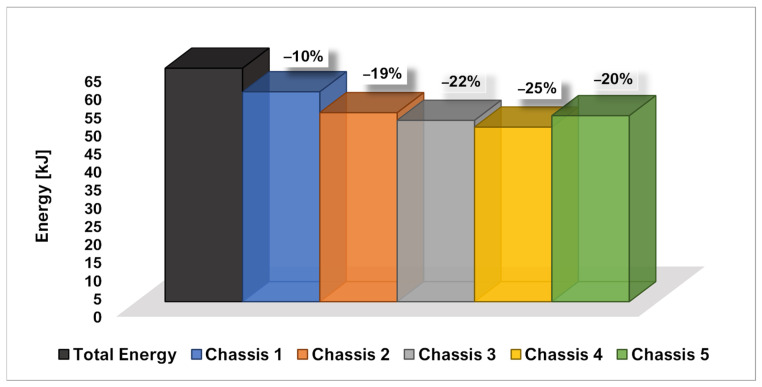
Comparison of energy detected in the safety cage.

**Figure 13 polymers-14-04163-f013:**
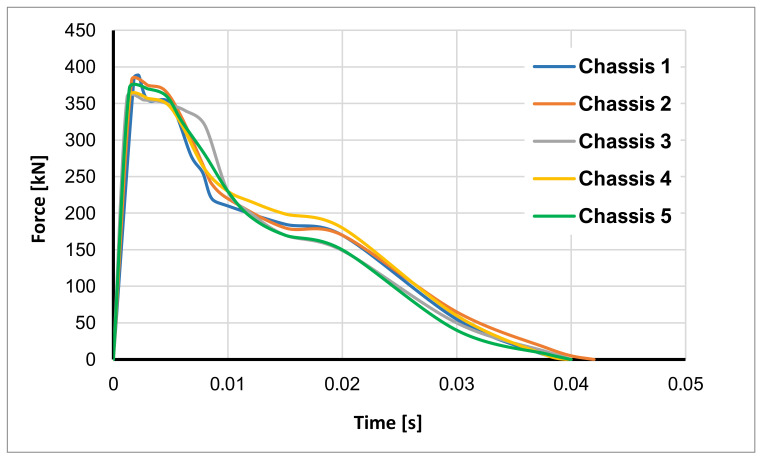
Force-time graph.

**Figure 14 polymers-14-04163-f014:**
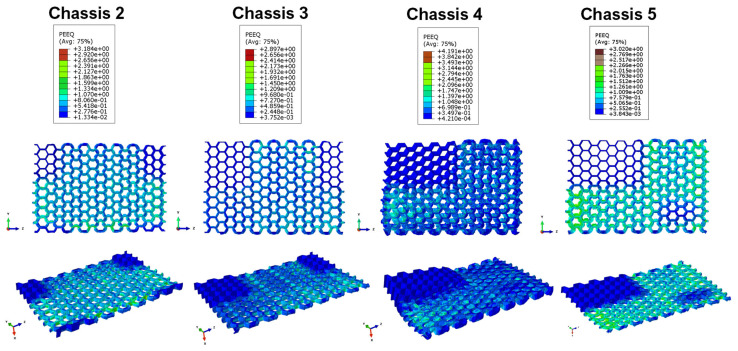
Equivalent Plastic Strain of the shock absorbers’ core.

**Figure 15 polymers-14-04163-f015:**
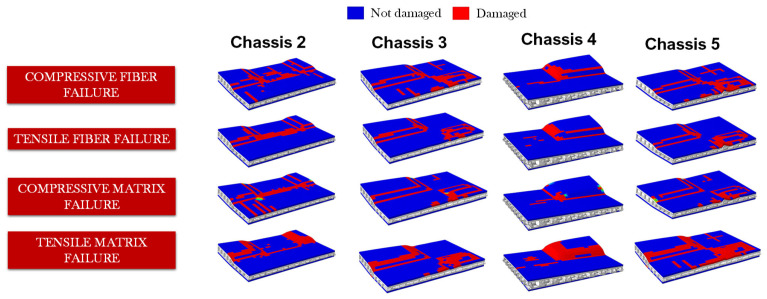
Comparison of intralaminar damages in the upper absorbers skin.

**Figure 16 polymers-14-04163-f016:**
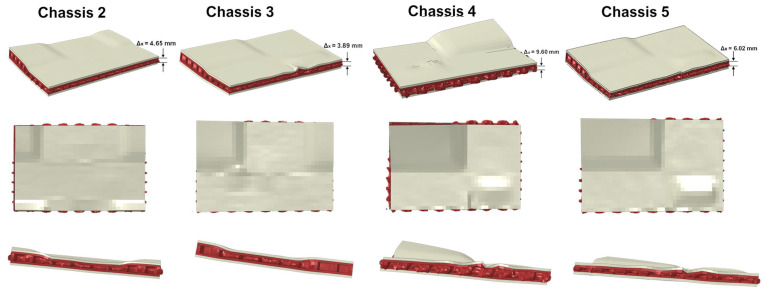
Shock absorbers’ deformed shape.

**Table 1 polymers-14-04163-t001:** Mechanical properties of IM7/977-2 composite material.

Property	Value
Density [kg/m^3^]	1580
E_1_ [MPa]	153,050.00
E_2_ = E_3_ [MPa]	10,300.00
G_12_ = G_13_ [MPa]	6000.00
G_23_ [MPa]	3700.00
ν_12_ = ν_13_	0.30
ν_23_	0.40
Longitudinal Tensile Strength [MPa]	1250.00
Longitudinal Compressive Strength [MPa]	850.00
Transverse Tensile Strength [MPa]	65.00
Transverse Compressive Strength [MPa]	200.00
Longitudinal Shear Strength [MPa]	75.00
Transverse Shear Strength [MPa]	35.00
Longitudinal Tensile Fracture Energy [kJ/m^2^]	15.00
Longitudinal Compressive Fracture Energy [kJ/m^2^]	7.00
Transverse Tensile Fracture Energy [kJ/m^2^]	0.50
Transverse Compressive Fracture Energy [kJ/m^2^]	4.00

**Table 2 polymers-14-04163-t002:** Mechanical properties of polypropylene [17].

Property	Value
Young’s modulus [MPa]	1620.00
Density [kg/m^3^]	1090
Poisson’s ratio	0.35
Tensile strength (at breakage) [MPa]	20
Elongation at breakage [%]	16

**Table 3 polymers-14-04163-t003:** Mechanical properties of ASTM A36 steel used for the chassis structure.

Property	Value
Density [kg/m^3^]	7890
Young’s Modulus [MPa]	200,000.00
Poisson’s ratio	0.26
**Plastic (Nominal)**	
A [MPa]	286.10
B [MPa]	500.10
η	0.2282
m	0
Melting Temperature	0
Transition Temperature	0
**Rate dependent (Nominal)**	
C	0.0171
${\dot{\bar{\varepsilon }}}_{0}$	1
**Damage evolution**	
d_1_	0.403
d_2_	1.107
d_3_	−1.899
d_4_	0.00961
d_5_	0.3
Reference Strain Rate	1

**Table 4 polymers-14-04163-t004:** Results summary.

Configuration	Mass [kg]	Total Absorbed Energy [kJ]	SEA [J/kg]	Peak Force Reduction [kN]	Maximum Out-of-Plane Displacement [mm]	Peak Stress on the Chassis [MPa]
*Chassis 2*	0.264	12.26	46.4	−7	4.65	720.0
*Chassis 3*	0.792	14.38	18.1	−30	3.89	652.9
*Chassis 4*	0.792	16.24	20.5	−29	9.60	604.1
*Chassis 5*	0.528	13.09	24.8	−18	6.02	670.6

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
