# Peer review of "On the Crashworthiness Behaviour of Innovative Sandwich Shock Absorbers"

_polymers, 2022, doi:10.3390/polym14194163_

Round 1

Reviewer 1 Report

In this article, the efficiency of additively manufactured shock-absorbers with composite material is assessed using numerical crash simulations. The authors stated that the 3D printing process produced high strength to mass and stiffness to mass sandwich structures, which resulted in an efficient damping system.

Major comments: The paper has an exhaustive discussion on mathematical equations and simulation work, which is already well known to the community. Moreover, the numerical work is not supported by any kind of experimental work in this article. 

The novelty of the work is the investigation of additively manufactured sandwich structure behavior under crash loadings. Applications of these additively manufactured structures can be further discussed for a chassis design. It is a good idea to consider a simple experimental case to validate their numerical results. It means that 3D print sandwich structures and conduct impact testing, and then validate the numerical work with experimental results.

Minor comments:

1. Introduction:

2. Theoretical background

2.1. Explicit integration Scheme adopted for the crash simulations

2.1 section is very elaborative, especially the first 3 paragraphs of the section. Write them concisely this section,

More importantly, In Section 2,  all mathematical equations can be moved to Appendix. discuss only essential parameters used in the simulation work, and the remaining equations can be moved to the Appendix section.

2.2. Intralaminar damage mechanisms

Can you explain the reason for the selection of Hashin failure criteria for damage modeling? Does the Hashin failure criteria represent a continuous damage model (Figure -1)?

Can you explain the damage evolution equation (12), what does K mean?

Also, explain the meaning of delta in equation 13 to calculate damage parameter (d). How do you calculate delta values for equation 13.

3. Results

In Figure 3, c and d, Can you indicate where shock absorbers are placed on the chassis? 

Author Response

Reviewer #1: 

In this article, the efficiency of additively manufactured shock-absorbers with composite material is assessed using numerical crash simulations. The authors stated that the 3D printing process produced high strength to mass and stiffness to mass sandwich structures, which resulted in an efficient damping system.

COMMENTS

  1. The paper has an exhaustive discussion on mathematical equations and simulation work, which is already well known to the community. Moreover, the numerical work is not supported by any kind of experimental work in this article. The novelty of the work is the investigation of additively manufactured sandwich structure behavior under crash loadings. Applications of these additively manufactured structures can be further discussed for a chassis design. It is a good idea to consider a simple experimental case to validate their numerical results. It means that 3D print sandwich structures and conduct impact testing, and then validate the numerical work with experimental results.
  2. The authors would like to thank the reviewer for this comment, as he pointed out that the aim of the work was unclear. Indeed, the present work is an automotive application test of additively manufactured shock-absorbing structures to increase the passive safety of commercial minibuses. The impact behaviour of the employed shock-absorbing structures has already been assessed by the authors in a previous work [15] by means of numerical analyses and experimental mechanical tests performed according to ASTM D7136. The authors updated the aim of the work and reported this information in the text with the following sentence:

An applicative test case in automotive field of this new structural shock absorbers’ concept is proposed in this work. In particular, this work is aimed to quantify the benefit of introducing several shock absorber systems, designed for Additive Manufacturing (DfAM) in key locations of a minibus chassis.

The dynamic response of the introduced absorber configurations has already been assessed both numerically and experimentally in [15,16]. Indeed, the absorber geometry is the same reported in [15]. In particular, it consists of a sandwich configuration with Polypropylene (PP) core and internal skin and carbon fibre composite (CFRP) external coating. This allows to combine the flexibility of the polymeric component with the lightness and strength of the external composite skins.”

  1. 2.1 section is very elaborative, especially the first 3 paragraphs of the section. Write them concisely this section, more importantly, In Section 2, all mathematical equations can be moved to Appendix. discuss only essential parameters used in the simulation work, and the remaining equations can be moved to the Appendix section.
  2. The reviewer provided a very interesting suggestion and the section 2 has been rearranged. In Section 2 an indication of the theoretical models considered for the evolution of material behavior was given, together with the required mechanical properties implemented in the software, while a description of the theoretical models with all the mathematical equations has been moved to the Appendix A section.

  1. Can you explain the reason for the selection of Hashin failure criteria for damage modelling? Does the Hashin failure criteria represent a continuous damage model (Figure 1)?
  2. The Hashin failure criteria has been employed in the investigation and implemented in the software as thei are based on Continuum Damage Mechanics formulation. The considered approach allows to better examine the damages evolution and identify the breakage of fibres and matrix under tensile and compressive loading. This point was better explained in the Section 2 by adding the following sentences:

“In particular, the onset and evolution of the intralaminar damage mechanisms of the shock absorber’s composite skins component has been assessed by using the Hashin’s criteria, based on a Continuous Damage Mechanics (CDM) formulation; this approach allows to identify the breakage of fibres and matrix and investigate the damages evolution under tensile and compressive loading.”

  1. Can you explain the damage evolution equation (12), what does K mean?
  2. In the equation (12), K represents the material stiffness in the undamaged condition. Authors added in the text specific information on the K parameter by introducing the following sentence:

“In this equation, K represent the material stiffness in the undamaged condition”.

  1. Also, explain the meaning of delta in equation 13 to calculate damage parameter (d). How do you calculate delta values for equation 13.
  2. Delta represents the equivalent displacement for an applied strain referred to a characteristic finite element length LC in several loading conditions during the damage evolution. Authors added in the text specific information on the delta parameter and on its evaluation by introducing the following sentence:

 is the equivalent displacement for an applied strain referred to a characteristic finite element length LC,  is the initial equivalent displacement at which the considered failure criteria is met, and  is the displacement at which the material is completely damaged.

An explanation for the calculation of delta values has been added by means of equations (14) and (15).

  1. In Figure 3, c and d, can you indicate where shock absorbers are placed on the chassis?
  2. After rearrangement of the section 2, Figure 3 became Figure 1. In Figure 1c, the chassis 3 configuration is characterized by a total of six devices placed in the corners and in the middle of the structure’s frontal section, while in Figure 1d the chassis 4 configuration is characterized by 4 absorbers placed only in the corners of the structure’s frontal section. This point was better explained in the Section 3 by modifying the sentences as follow:

“In chassis 3, the shock absorption system of chassis 2 has been extended, assuming a total of six devices placed in the corners and in the middle of the structure’s frontal section, each one with a sandwich height of 15 mm. The total shock absorbing system mass is 0.792 kg (Figure 1C);

Chassis 4 includes a damping system with the same mass as the one introduced in Chassis 3, but composed of only 4 absorbers. In this configuration, the shock absorbers are placed only in the corners of the structure’s frontal section. However, each absorber is 23 mm high. The mass of this damping system is 0.792 kg (Figure 1D).”

A detailed representation of the shock absorbers positioning was already provided in the Figure 4 b and c.

Reviewer 2 Report

The manuscript discussed on the Crashworthiness Behaviour of Additive Manufactured Sandwich Shock Absorbers. The comments for the manuscript are as follows.

1. The caption of the Table always comes above it. Put caption of Table 1 above it.

2. Give detailed information about the assumption being considered while FE model.

3. Introduction part should be enriched with pertinent literature. The work done by the past researchers should be clearly highlighted and how authors have come up with the solution.

4. In abstract section, outline the novelty of the work. 

5. Conclusion is too simple and I suggest to write the major outcomes of the study with future prospects.

6. I recommend to annotate the figure with suitable labelling for better insight.

7. Give more detailed information about Figure 17 and Figure 18 along with technical interpretation.

8. . The dynamic response of the absorber configurations should be elaborated more in the introduction section.

Author Response

Reviewer #2:

The manuscript discussed on the Crashworthiness Behaviour of Additive Manufactured Sandwich Shock Absorbers.

COMMENTS

  1. The caption of the Table always comes above it. Put caption of Table 1 above it.
  2. Done. The captions of all the tables have been placed above them, as the reviewer suggested.

  1. Give detailed information about the assumption being considered while FE model.
  2. Section 3 describing the FE model has been revised by adding additional information. Thank you for your suggestion.

  1. Introduction part should be enriched with pertinent literature. The work done by the past researchers should be clearly highlighted and how authors have come up with the solution.
  2. Done. As the reviewer suggested, the introduction part has been enriched with more literature examples.

  1. In abstract section, outline the novelty of the work.
  2. A clearer identification of the novelty of the work was introduced in the abstract section. Authors added in the abstract section specific information on the novelty of the work by introducing the following sentence:

“The use of high-efficiency additively manufactured sandwich shock absorbers has been demonstrated as an effective way to improve the passive safety of passengers achieving a reduction of the peak accelerations and energy absorbed in the safety cage of the chassis’s structure.”

  1. Conclusion is too simple and I suggest to write the major outcomes of the study with future prospects.
  2. Done. The conclusive section has been deeply revised, highlighting the major outcomes of the work. Below is the revised version of the conclusion:

“This paper assesses the effectiveness, in terms of energy dissipation, of PP/CFRP composite shock absorbers designed for Additive Manufacturing, under realistic oper-ating conditions.

In particular, the effects on the maximum von Mises stress reached on the chassis of minibuses subjected to a 50 km/h crash test with and without the integration of dif-ferent damping systems are compared. The work shows that the introduction of dampers can reduce the maximum stress on the chassis by up to 24%. Despite in all damping system configurations the stress fields exceeds the allowable stress limit val-ue, the employment of the shock absorber devices is able to significantly reduce the area characterized by values exceeding the maximum limit. Furthermore, the effec-tiveness of the energy dissipation systems has been assessed by comparing the energy recorded in the safety cage, the force-time graphs and the damage mechanisms of the composite shock absorbers.

The comparison of the energy recorded in the safety cage demonstrated that the introduction of the shock absorber devices can reduce the energy transferred in the safety cage up to 25%. The performed analyses showed that the 90% of the total energy was transferred to the safety cage in the case without shock absorber devices, while the employment of only two shock absorbers in Chassis 2 configuration demonstrated able to reduce the total transferred energy to 81%. A maximum reduction of the total ener-gy transferred to the safety cage to 75% has been achieved in the Chassis 4 configura-tion by means of four thicker devices.

Moreover, the comparison of the curves on the force-time graph showed that the introduction of the shock absorber devices allowed a reduction in the maximum peak of the reaction force up to 30 kN. The ability of the damping systems to dissipate im-pact energy by means of plastic deformation of the shock absorbers’ core and in-tralaminar cracking of the composite skins significantly improve the crashworthiness of the structure and passive safety of passengers, resulting in a reduced perception of impact acceleration by the occupants. The comparisons of the equivalent plastic strains for the shock absorbers’ cores and of the intralaminar damages for the compo-site skins demonstrated that both components effectively contribute to the dissipation of the impact energy.”

  1. I recommend to annotate the figure with suitable labelling for better insight.
  2. As suggested by the reviewer, figures have been annotated in the text in cursive to improve identification and insight.

  1. Give more detailed information about Figure 17 and Figure 18 along with technical interpretation.
  2. Done. After a rearrangement of the Section 2, Figure 17 and Figure 18 became Figure 15 and Figure 16. Authors added further information about the figures by introducing the following sentence:

“In Figure 15, a comparison of the intralaminar damages in the upper skin of a representative absorber of each shock-absorbing system is presented. The adopted failure criteria allowed to investigate the specific intralaminar failure modes of the composite components. In Chassis 2 and Chassis 3 configurations, the extension of all damages is quite similar for all the failure modes denoting a massive contribution of the onset and propagation of intralaminar damages in the energy absorption of the shock absorber. In Chassis 4, the extension of all damages is reduced with respect to both Chassis 2 and Chassis 3 configuration, denoting a reduced contribution of the intralaminar damages in the energy absorption of the shock absorber, while a major contribution can be assumed by the thicker plastic core.

Finally, in Figure 1618 the behaviour of a representative absorber of each shock-absorbing system is compared in terms of deformed shape. The comparison al-lowed the investigation of the deformation values in the X direction, which is the direction of the crash test, in order to better understand the influence and the contribution of the plastic core in the energy absorption capability of the different shock absorber configurations.”

  1. The dynamic response of the absorber configurations should be elaborated more in the introduction section.
  2. Done. The dynamic response of the absorbers was already presented and assessed by the authors in a previous work [15] through numerical and experimental analyses. However, a discussion about the dynamic response of the absorbers has been introduced in the Introduction section, as the reviewer suggested. This point was better explained by adding the following sentences:

“An applicative test case in automotive field of this new structural shock absorbers’ concept is proposed in this work. In particular, this work is aimed to quantify the benefit of introducing several shock absorber systems, designed for Additive Manufacturing (DfAM) in key locations of a minibus chassis.

The dynamic response of the introduced absorber configurations has already been assessed both numerically and experimentally in [15,16]. Indeed, the absorber geometry is the same reported in [15]. In particular, it consists of a sandwich configuration with Polypropylene (PP) core and internal skin and carbon fibre composite (CFRP) ex-ternal coating. This allows to combine the flexibility of the polymeric component with the lightness and strength of the external composite skins. All components in the shock absorber device fulfil a specific role during the impact phenomena: the Polypropylene core can easily dissipate the impact energy through plasticisation mechanisms, the composite skin reinforce the sandwich structure and contribute to the energy dissipation through onset and propagation of intralaminar damages, while the PP skin allows a better adhesion between the PP core and the composite skin, avoiding delaminations. The employment of a thinner and a thicker core allows the investigation of the influence of the PP core thickness on the energy absorption capabilities of the device.”

Reviewer 3 Report

Authors compared different polymer/composite (Polypropylene/Composite Fibres Reinforced Polymers – PP/CFRP) shock absorbers and tried to find out the best one which can be integrated in key locations of a minibus chassis via numerical crash simulations. This is an interesting work and has a good application prospect in the field of vehicle transportation safety. But frankly speaking, this work has nothing to do with additive manufacturing, even the title is “Additive Manufactured Sandwich Shock Absorbers”.

In the Theoretical background part, authors spent a lot of time explaining too much basic knowledge, many of which can be easily found in textbooks, such as Figure 2, stress-strain relationship curve. I suggest they cut this part on a large scale. If they think these are necessary, then can be put into the supporting documents as attachments.

On the other hand, the explanatory texts below pictures (illustration) are too simple. For example, Figure 7 needs to explain what A and B figures are in detail.

The selection of samples is unclear. Since chassis 3 is obviously better than chassis 2, why don't use the arrangement of chassis 3 and continue to increase thickness of absorbers. 

Author Response

Reviewer #3:

Authors compared different polymer/composite (Polypropylene/Composite Fibres Reinforced Polymers – PP/CFRP) shock absorbers and tried to find out the best one which can be integrated in key locations of a minibus chassis via numerical crash simulations.

COMMENTS

  1. This is an interesting work and has a good application prospect in the field of vehicle transportation safety. But frankly speaking, this work has nothing to do with additive manufacturing, even the title is “Additive Manufactured Sandwich Shock Absorbers”.
  2. Thanks to the reviewer's suggestion, it was decided to shift the focus of the work to the effectiveness of the proposed solutions in this application case. Therefore, the title has been changed by replacing the term 'additive manufactured' with 'innovative'.

  1. In the Theoretical background part, authors spent a lot of time explaining too much basic knowledge, many of which can be easily found in textbooks, such as Figure 2, stress-strain relationship curve. I suggest they cut this part on a large scale. If they think these are necessary, then can be put into the supporting documents as attachments.
  2. The reviewer provided a very interesting suggestion and the section 2 has been rearranged. In Section 2 an indication of the theoretical models considered for the evolution of material behavior was given, together with the required mechanical properties implemented in the software, while a description of the theoretical models with all the mathematical equations has been moved to the Appendix A section.

  1. On the other hand, the explanatory texts below pictures (illustration) are too simple. For example, Figure 7 needs to explain what A and B figures are in detail.
  2. The captioning of the figures has been revised. In particular, the caption of the indicated figure has been revised as follows:

Finite Element discretization. A) Computational grid of the chassis with SC8R elements; B) Com-putational grid of the shock absorber with SC8R elements (in green) and C3D8R elements (in grey).”

  1. The selection of samples is unclear. Since chassis 3 is obviously better than chassis 2, why don't use the arrangement of chassis 3 and continue to increase thickness of absorbers.
  2. The reviewer's note is right and allowed the authors to realise that the selection of samples had not been well described. It was decided to use the shock-absorbing configuration proposed in Chassis 4 as this, being composed mainly of polypropylene which is a thermoplastic polymer, is more economical and environmentally friendly than that of Chassis 3. This has been better explained in the text by inserting the following sentence:

“This configuration was considered because, differently from the Chassis 3 config-uration, it is mainly composed by polypropylene (a thermoplastic polymer) and this makes it cheaper than the Chassis 3 configuration and more environmentally friendly.”

Round 2

Reviewer 2 Report

Authors have included the recommended suggestions, therefore, can be considered for publication. 

Author Response

The authors would like to thanks the reviewer for his precious suggestions

Reviewer 3 Report

Authors have adjusted the structure of the paper and now it reads more smoothly. But I have some suggestions. 1. The attachment (Appendix ) shall be a separate supporting document and need not be included in the main text. 2. Although authors explain the logic of Chassis 4. I still donot think it is suitable to choose. Since authors select Chassis 4 from the economic point of view, it is necessary to first compare the results of using Chassis 4 with same thickness of Chassis 3. Then, study the thickness of the polymer is changed based on the selection of Chassis 3 and Chassis 4. In addition, the cost data of the price is attached to find the optimal solution. Because if economic factors are to be considered, it may be meaningful to choose Chassis  3 and make the polymer thinner.

Author Response

Reviewer #3: 

Authors have adjusted the structure of the paper and now it reads more smoothly. But I have some suggestions:

 COMMENT N°1

  • The attachment (Appendix ) shall be a separate supporting document and need not be included in the main text.
  • The authors would like to thank the reviewer for his precious comments. The appendix has been attached as a separate supporting document.

COMMENT N°2

  • Although authors explain the logic of Chassis 4. I still do not think it is suitable to choose. Since authors select Chassis 4 from the economic point of view, it is necessary to first compare the results of using Chassis 4 with same thickness of Chassis 3. Then, study the thickness of the polymer is changed based on the selection of Chassis 3 and Chassis 4. In addition, the cost data of the price is attached to find the optimal solution. Because if economic factors are to be considered, it may be meaningful to choose Chassis 3 and make the polymer thinner.
  • Authors understand the concern of the reviewer, hence a new configuration has been added, named Chassis 5, designed starting from Chassis 4 arrangement, but adopting the same core thickness of Chassis 3. The results from these configurations have been cross-compared and critically discussed. To reflect this aspect, section 4 in the revised version of the manuscript, has been significantly modified.